# Pipette-Free and Fully Integrated Paper Device Employing DNA Extraction, Isothermal Amplification, and Carmoisine-Based Colorimetric Detection for Determining Infectious Pathogens

**DOI:** 10.3390/s23229112

**Published:** 2023-11-11

**Authors:** Hanh An Nguyen, Nae Yoon Lee

**Affiliations:** Department of BioNano Technology, Gachon University, 1342 Seongnam-daero, Sujeong-gu, Seongnam-si 13120, Republic of Korea; hanhannguyen96@gmail.com

**Keywords:** Pipette-free and fully integrated paper device, carmoisine-based colorimetric detection, *Enterococcus faecium*, point-of-care testing

## Abstract

A pipette-free and fully integrated device that can be used to accurately recognize the presence of infectious pathogens is an important and useful tool in point-of-care testing, particularly when aiming to decrease the unpredictable threats posed by disease outbreak. In this study, a paper device is developed to integrate the three main processes required for detecting infectious pathogens, including DNA extraction, loop-mediated isothermal amplification (LAMP), and detection. All key reagents, including sodium dodecyl sulfate (SDS), NaOH, LAMP reagents, and carmoisine, are placed on the paper device. The paper device is operated simply via sliding and folding without using any bulky equipment, and the results can be directly observed by the naked eye. The optimized concentrations of sodium dodecyl sulfate (SDS), sodium hydroxide (NaOH), and carmoisine were found to be 0.1%, 0.1 M, and 0.5 mg/mL, respectively. The paper device was used to detect *Enterococcus faecium* at concentrations as low as 10^2^ CFU/mL within 60 min. Also, *E. faecium* spiked in milk was successfully detected using the paper device, demonstrating the feasible application in real sample analysis.

## 1. Introduction

In recent years, infectious pathogens have become an increasing challenge and pose an unpredictable threat to public health. In response, increasing research efforts have been put into the development of point-of-care testing (POCT) in order to discover effective solutions. POCT, also called near-patient testing, encompasses any diagnostic tests that can be performed outside the centralized laboratory [1,2,3]. POCT provides not only critical solutions for controlling the spread of infectious pathogens, but also key strategies for timely treatment. Current POCT strategies for infectious pathogens can be delineated into two main categories. The first category encompasses immunological assays, which detect infectious pathogens based on the specific interaction between the antibodies of the patients and the antigens from the pathogens [4,5,6]. However, immunological assays have been associated with some false negative results caused by the delay between infection and immune initiation. The second category encompasses molecular diagnostics, which is based on the specific genetic information of infectious pathogens; therefore, the issue of false negative results can be prevented [7,8,9].

Molecular diagnostics involves three main steps: DNA extraction, amplification, and detection. In DNA extraction, the sodium dodecyl sulfate (SDS)-based method serves as one of the most effective methods for breaking microbial membranes in order to release DNA. The SDS-based DNA extraction method has been widely employed for a wide range of environmental samples, plant and animal tissues, and clinical specimens. The principle of this method relies on using SDS to break organisms’ membranes, using phenol/chloroform/isoamyl alcohol to degrade non-DNA molecules, and using isopropanol to precipitate DNA [10,11,12]. However, the SDS-based method is not considered to be suitable for POCT because it is a complicated procedure with multiple steps that requires bulky equipment. Therefore, the SDS-based method needs to be modified in order to simplify the process, reducing the number of steps and omitting bulky apparatuses.

For DNA amplification, isothermal techniques have recently gathered much attention from researchers owing to their rapidity, cost effectiveness, ease of use, and ability to be performed under constant temperature. Some common isothermal techniques include loop-mediated isothermal amplification (LAMP) [13,14,15], rolling circle amplification (RCA) [16,17], nucleic acid sequence-based amplification (NASBA) [18,19], and recombinase polymerase amplification (RPA) [20,21]. Among these listed isothermal techniques, LAMP serves as the most promising approach because it requires less enzyme compared to RPA. In addition, LAMP does not require multiple steps, as RCA does; instead, LAMP can be performed in a single step. Compared with NASBA, LAMP shows relatively higher specificity because LAMP uses more pairs of primers. Due to the advantages listed above, LAMP satisfies the requirements for POCT and exhibits great potential for use in resource-limited settings.

For the readout of LAMP results, several techniques have been developed, such as electrochemical analysis, gel electrophoresis, and fluorescence-based detection [22,23,24]. However, the listed techniques require external electric power, bulky apparatuses, and multiple steps. To overcome the limitations of these techniques, colorimetric detection has been developed. Colorimetric detection uses a dye to indicate the presence or absence of DNA amplicons through color change. One of the most obvious advantages of colorimetric detection is that the results can be observed by the naked eye. In addition, colorimetric detection has a simple operation, high sensitivity, great selectivity, and a low cost [25,26,27,28,29]. Therefore, colorimetric detection can fulfill the requirements for application in POCT.

The concept of a paper device was introduced for the first time in a landmark report by Whitesides’ group in 2007, and since then a large number of paper devices have been developed for application in POCT [30,31,32,33]. Compared with other materials, devices made from paper present unique advantages, such as an economical cost, simple fabrication, and the use of eco-friendly materials. Therefore, in this study, we aim to develop a pipette-free device using paper-based materials that integrates all key processes involved in pathogen detection, including DNA extraction, LAMP amplification, and colorimetric detection.

## 2. Materials and Methods

### 2.1. Materials

Hydrogen peroxide (H_2_O_2_), copper(II) sulfate pentahydrate (CuSO_4_·5H_2_O), carmoisine, and sodium hydroxide (NaOH) were obtained from Sigma-Aldrich (St. Louis, MO, USA). The LAMP reagents, including *Bst* 2.0 WarmStart DNA polymerase, 25 mM of MgCl_2_, 10× isothermal amplification buffer and dNTP mix were purchased from New England Biolabs (Ipswich, MA, USA). For chip fabrication, Whatman grade 2 filter paper was purchased from GE Healthcare Life Sciences (Chicago, IL, USA). Felts and hard paper were bought from a local market (Seoul, Republic of Korea). The polydimethylsiloxane (PDMS) prepolymer (Sylgard 184) and curing agent were provided by Dow Corning. Luria-Bertani low-salt broth was purchased from MBcell (Seoul, Republic of Korea). Agarose powder, loading STAR, and DNA ladder (100 bp) were purchased from BioShop (Burlington, ON, Canada), Dyne Bio (Seongnam, South Korea), and Takara (Shiga, Japan), respectively. To detect the target bands, the Gel Doc EZ System (Bio-Rad, Hercules, CA, USA) was used.

### 2.2. Preparation of Bacterial Samples and DNA Extraction

Bacterial samples were prepared by culturing *Enterococcus faecium* (*E. faecium*, ATCC BAA-2127) in 5 mL of Brain Heart Infusion (BHI) media. *E. faecium* culture was incubated at 37 °C for 16 h and constantly shaken at 200 rpm in a shaking incubator. For DNA extraction, 10 µL of *E. faecium* culture was loaded into a 10 µL lysis buffer that contained 0.1 M NaOH, sodium dodecyl sulfate (SDS) 0.1%, and 100 mg/mL of NaCl. The lysis reaction was performed at room temperature for 5 min. A Whatman paper disc (1.5 mm in diameter) was then dipped into lysed *E. faecium* to collect the DNA released from *E. faecium*. Next, the washing step was performed by dipping DNA-binding paper disc into the Tris-EDTA (TE) buffer to remove LAMP inhibitors.

### 2.3. LAMP Reaction

To detect *E. faecium*, the *esp* gene, which is an essential gene contributing to biofilm formation, was selected as a target. A primer set for amplifying the *esp* gene was designed using PrimerExplorer version 5 software. The sequences of primers are shown in Table 1. Information about the *esp* sequence was found in the National Library of Medicine. For LAMP reaction, a 25 µL mixture of LAMP reagents contained isothermal amplification buffer, 1 mM of dNTPs, 2 mM of MgSO_4_, 3.2 µM of each inner primer (FIP and BIP), 0.4 µM of each outer primer (F3 and B3), 1.6 µM of each loop primer (LB), and 0.1 U/µL of *Bst* 2.0 WarmStart DNA polymerase. Carmmoisine (0.5 mg/mL) was added into the LAMP mixture. LAMP reactions were performed at 65 °C for 45 min. To examine the reliability of the experiments, a negative control not containing the DNA template and a positive control containing the DNA template were also tested.

### 2.4. Optimization of Carmoisine-Based Colorimetric Detection

To monitor the LAMP results, 100 mM of CuSO_4_, 0.35% H_2_O_2_, and LAMP amplicons containing 0.5 mg/mL of carmoisine were mixed at a volume ratio of 1:2:10. The reaction occurred at room temperature for 5 min. Through colorimetric detection, the presence or absence of pathogens could be observed by the naked eyes. The difference in color was determined according to the ratio of mean gray values in the negative samples and tested samples using ImageJ 1.52a software. The higher the ratio, the greater the difference in color between the negative and tested samples.

### 2.5. Fabrication of Paper Device

The paper device consisted of three main Pads (Pad 1, 2 and 3), as shown in Figure 1. Pad 1 was fabricated from a hard paper sheet (50 × 50 mm^2^). On Pad 1, eight pillars made from PDMS-coated felt discs were arranged into two lines. Four Whatman paper discs (3 mm in diameter) were attached to the outer line pillars using double-sided tape. Four CuSO_4_-coated Whatman paper discs and four H_2_O_2_-coated Whatman paper discs were stacked on the inner line of the pillars using double-sided tape. Pad 2 was fabricated from hard paper (38 × 136 mm^2^), and had four holes that were cut using a laser cutting machine. Pad 1 was attached to Pad 2 using double-sided tape. Then, Pad 2 was folded to form a hollow box that could move Pad 1 via a simple sliding motion. Pad 3 was fabricated from PDMS-coated felt (48 × 50 mm^2^, 2 mm thickness) with twelve holes, and was stacked onto a PDMS-coated Whatman paper (48 × 50 mm^2^) with no holes, creating twelve chambers aligned into 3 lines. The first and second lines of chambers contained the reagents needed for DNA extraction, which were lysis buffer (NaOH, SDS, and NaCl) and washing buffer (TE buffer), respectively. The third line of chambers contained the reagents needed for DNA amplification and detection, which were the LAMP reagents and carmoisine.

The PDMS coating process was performed following the protocol used in our previous studies [14]. First, the PDMS prepolymer was thoroughly mixed with a curing agent at a ratio of 10:1 (*w*/*w*). Then, the mixture was placed in a vacuum oven for 20 min to remove air bubbles inside the mixture. After that, the mixture was brushed onto the paper. The PDMS-brushed paper was incubated at 80 °C for 6 h.

### 2.6. Operation of Paper Device

Figure 2 illustrates the operation of the paper device for pathogen detection. First, 10 µL of bacterial sample was loaded into the chambers containing lysis buffer. The lysis reaction was performed at room temperature for 5 min. Pad 1 was then folded in order to dip the Whatman paper discs on the outer pillars into the lysed samples to collect the DNA released from the bacterial pathogen. After that, Pad 1 was pulled up to open the chambers. Next, Pad 2 was slid forward to the second line of chambers, which contained the washing buffer. Pad 1 was then folded in order to dip the DNA-collected paper discs into the washing buffer for 5 min. To perform DNA amplification, the same pulling, sliding, and folding action was repeated to transport the DNA to the third line of chambers, which contained the LAMP reagents and carmoisine. The paper device was then incubated at 65 °C for 45 min on a hot plate. Double-sided tape was used to maintain the connection between Pad 1 and the third line of chambers. After the LAMP reaction, the colorimetric reagent-loaded paper discs attached to the inner pillars were dipped into the chambers. After 5 min, the results could be observed on the inner pillars.

## 3. Results and Discussion

### 3.1. DNA Extraction Performance

The NaOH and SDS concentrations were optimized in microtubes by testing ranges of concentrations from 0.1 M to 1 M and from 0.1% to 10%, respectively. Then, 10 µL of bacterial solution was directly loaded into the 10 µL of lysis buffer. The lysis reaction was performed at room temperature for 5 min. A Whatman paper disc (3 mm in diameter) was dipped inside the lysed bacterial solution to collect the released DNA. In this step, large molecules of DNA become rapidly entrapped inside the cellulose matrix of the Whatman paper and retained during the washing step, while small compounds including DNA amplification inhibitors such as NaOH and SDS are released into the washing buffer [34]. After washing, the DNA-binding Whatman paper disc was transferred into microtubes containing the LAMP reagents, followed by heat incubation at 65 °C for 45 min. The results were checked using gel electrophoresis. Negative samples (not containing bacterial solution) and positive samples (containing extracted DNA using Wizard^®^ Genomic DNA Purification Kit, Upper Coomera, Australia) were also included. As shown in Figure 3, two combinations were successfully used to extract DNA from the bacterial solution: 0.5 M NaOH/0.1% SDS and 0.1 M NaOH/0.1% SDS. In order to reduce the cost of the test, 0.1 M NaOH and 0.1% SDS—the lowest concentrations that could be efficiently used for DNA extraction—were employed for later experiments. The other seven combinations of NaOH and SDS could not be used to extract DNA because excess amounts of NaOH and SDS could damage DNA; therefore, a more efficient washing step is needed to eliminate their inhibitory effect in the subsequent process.

### 3.2. Optimization of Carmoisine-Based Colorimetric Detection

Carmoisine-based colorimetric detection was performed according to our previous study [14]. Carmoisine, also known as Azorubine, is classified as an azo dye composed of two naphthalene subunits joined via an azo linkage (–N=N–) [35]. To employ carmoisine for colorimetric detection, hydroxyl radicals generated by the reaction between CuSO_4_ and H_2_O_2_ were used to degrade the azo linkage of carmoisine, leading to the decolorization of carmoisine from purple to colorless. However, the presence of target DNA interferes with carmoisine decolorization by reacting with hydroxyl radicals at the hydrogen atoms of the deoxyribose, resulting in the purple color of carmoisine being retained (Figure 4a) [36].

As the results in Figure 4b,c show, the difference in color between the negative and positive samples decreases over time because carmoisine keeps reacting with the hydroxyl radical generated by CuSO_4_ and H_2_O_2_. To overcome this problem, a Whatman paper disc was dipped into the mixture after the colorimetric reaction. Then, the dipped paper disc was taken out to stop the decolorization. The color on the paper disc was also observed at 0, 5, 10, 15, and 20 min. The results shown in Figure 4d,e demonstrate the stability of the color on the paper discs over time. The difference in color on the paper disc between the negative and positive samples remains stable from 0 to 20 min.

To evaluate the compatibility of carmoisine in the LAMP reaction, DNA amplification using the LAMP reaction was performed under the presence of carmoisine in the LAMP mixture. Carmoisine was mixed with the LAMP mixture in microtubes, followed by heat incubation at 65 °C for 45 min. After the LAMP reaction, H_2_O_2_-coated and CuSO_4_-coated Whatman paper discs were placed inside the microtubes containing LAMP amplicons to activate carmoisine-based colorimetric detection. After 5 min at room temperature, the two paper discs (H_2_O_2_-coated and CuSO_4_-coated paper discs) were taken out and the color on the paper was observed. Three carmoisine concentrations (0.1, 0.5, and 1 mg/mL) were tested in order to determine the most optimized concentration that shows the greatest difference in color on the paper discs in the negative and positive samples. The difference in color on the paper discs was determined using the coefficient between the mean gray values in the negative and positive samples using ImageJ software. As the results in Figure 5a,b show, with 0.5 mg/mL of carmoisine concentration, the positive samples can be easily distinguished from the negative samples by the naked eye on both the H_2_O_2_-coated and CuSO_4_-coated paper discs. The gel electrophoresis results in Figure 5c confirm that carmoisine did not interfere with the LAMP reaction. By mixing carmoisine with LAMP reagents before the DNA amplification and by using reagent-coated paper discs, colorimetric detection could be simplified via the complete elimination of the pipetting steps.

### 3.3. Selectivity

The selectivity test was performed using *E. faecium* primers to amplify the DNA from *Staphylococcus aureus (S. aureus*). The colorimetric results are shown in Figure 6a. LAMP reactions were successfully performed only when *E. faecium* primers were used to amplify the DNA from *E. faecium*. When *E. faecium* primers were used to amplify *S. aureus*, the amplification was unsuccessful. The results were confirmed via the performance of gel electrophoresis, as shown in Figure 6b.

### 3.4. Sensitivity

To evaluate the performance of the paper device, its sensitivity was first examined via the testing of a range concentrations of *E. faecium*, from 10^1^ to 10^3^ CFU/mL. The paper device was operated as described in Section 2.6. Although the color difference indicating the presence and absence of LAMP amplicons could be observed directly on paper discs and inside the third line of chambers, the color in the liquid form decreased over time, as shown in Figure 4a. Therefore, the results were observed on the paper discs instead of in the liquid form inside the chambers. Figure 6c,d shows the results of the sensitivity test. The lowest *E. faecium* concentration that could be detected using the paper device was 10^2^ CFU/mL. The results were confirmed via gel electrophoresis, as shown in Figure 6e.

### 3.5. Real Sample Analysis

The paper device was used to detect *E. faecium* spiked in milk to evaluate the performance of the device for real samples. As shown in Figure 6f,g, the presence of *E. faecium* in the milk sample could be recognized through color change on the paper discs. The results were verified via gel electrophoresis, as shown in Figure 6h. By integrating the three processes required for molecular testing into the paper device, the multistep assay for pathogen detection can be made more user-friendly: (1) the pipetting steps are completely eliminated, (2) the paper device can be simply operated by sliding and folding, and (3) results can be observed directly by the naked eye.

## 4. Conclusions

In this study, a pipette-free paper device was fabricated in order to integrate SDS-based DNA extraction, LAMP amplification, and carmoisine-based colorimetric detection. The paper device contained all the reagents required for a molecular test, such as lysis buffer for DNA extraction, LAMP reagents for DNA amplification, and carmoisine, H_2_O_2_, and CuSO_4_ for colorimetric detection. Thus, this paper device could be used to perform all the reactions required for a molecular test to detect infectious pathogens, without using bulky laboratory equipment. The paper device was used to detect *E. faecium* at concentrations as low as 10^2^ CFU/mL within 60 min. Also, *E. faecium* spiked in milk was successfully detected using the paper device, demonstrating the feasibility of its application in real sample analysis. The paper device therefore serves as a promising tool for POCT, especially in resource-limited environments, for the following reasons: (1) it is simple to fabricate; (2) it can be simply operated via sliding and folding without using bulky equipment; and (3) the results can be observed by the naked eye. However, some limitations remain to be overcome in order to bring the developed paper device closer to end users. For example, further research should investigate multiplex detection. The selectivity of the *E. faecium* primer also needs to be examined using other *Enterococci* strains, such as *Enterococcus faecalis*, *Enterococcus cecorum*, and *Enterococcus durans*, which also cause opportunistic infection.

## Figures and Tables

**Figure 1 sensors-23-09112-f001:**
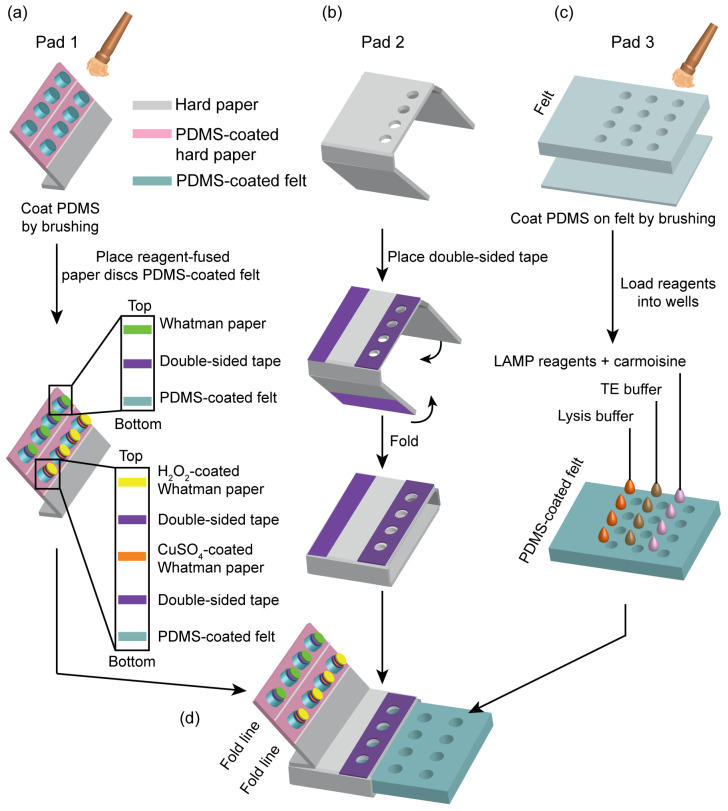
Overview of the paper device fabrication. (**a**) Pad 1 contains four outer pillars and four inner pillars that carry Whatman paper discs and reagent-containing Whatman paper discs for colorimetric detection, respectively. (**b**) Pad 2 is fabricated with the ability to be folded into a hollow box. (**c**) Pad 3 is made from a twelve-hole PDMS-coated felt attached on a Whatman paper of the same size with no holes. (**d**) Illustration of the paper device.

**Figure 2 sensors-23-09112-f002:**
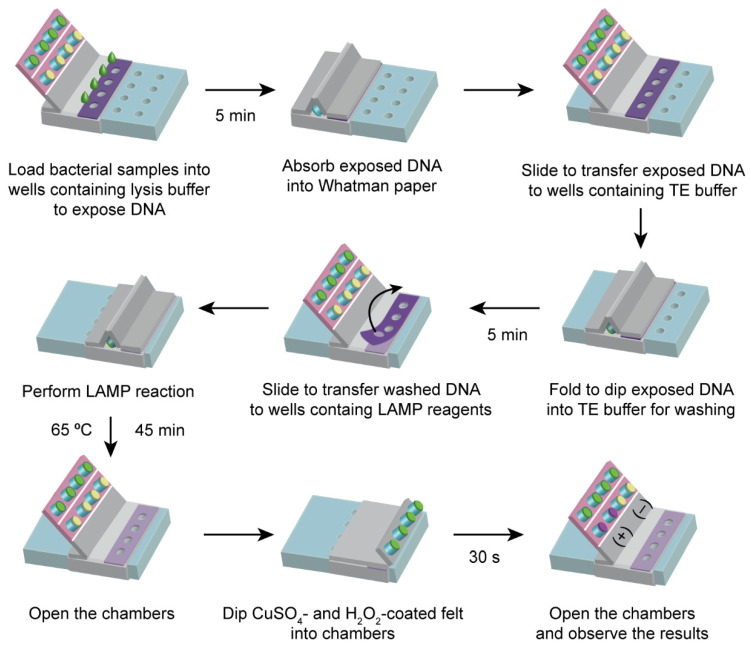
Operation of the paper device for DNA extraction, LAMP reaction, and colorimetric detection.

**Figure 3 sensors-23-09112-f003:**
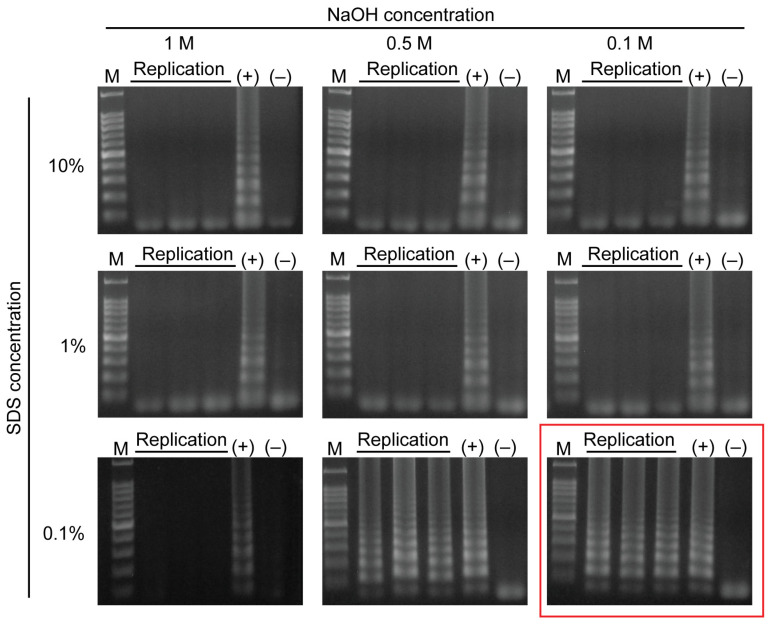
Optimization of NaOH (0.1 M, 0.5 M, and 1 M) and SDS (0.1%, 1%, and 10%) concentrations for DNA extraction. M represents the 100 bp DNA ladder. (+) represents samples containing extracted DNA using a commercial kit. (−) represents samples not containing bacteria.

**Figure 4 sensors-23-09112-f004:**
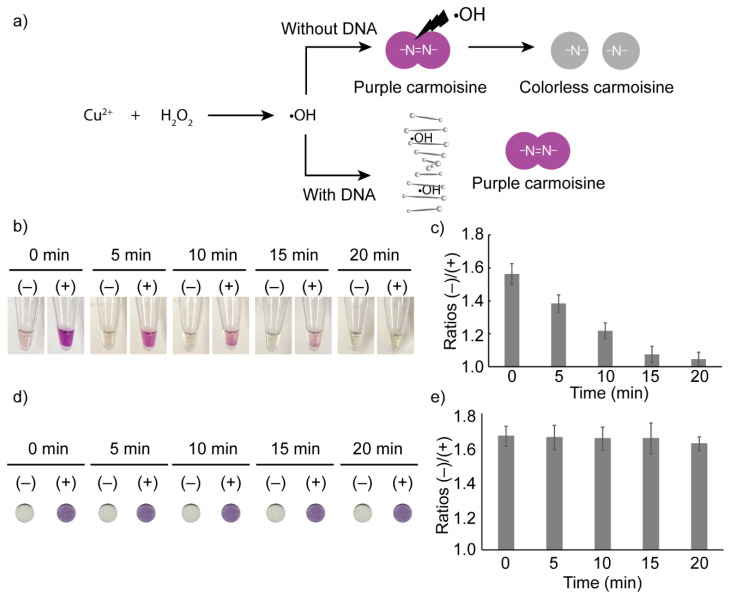
Optimization of carmoisine-based detection. (**a**) Principle of carmoisine-based detection. (**b**) Effect of time on the colorimetric detection occurring in liquid form. (**c**) Ratios (−)/(+) represent the ratios of the mean gray values of the negative and positive samples from 0 min to 20 min that were measured in liquid form. (**d**) Effect of time on color change on paper over time. (**e**) Ratios (−)/(+) represent the ratios of the mean gray values of the negative and positive samples that were measured on paper discs.

**Figure 5 sensors-23-09112-f005:**
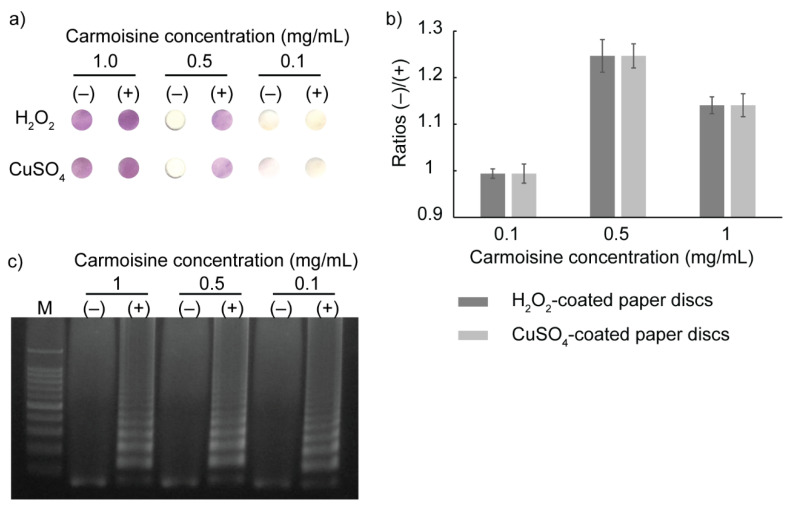
Compatibility of carmoisine-based colorimetric detection in LAMP reaction. (**a**) Optimization of carmoisine concentration. (−) samples not containing DNA. (+) samples containing DNA. (**b**) The ratios of mean gray values of negative and positive samples measured on paper discs. (**c**) Electrophoresis results showing the compatibility of carmoisine in the LAMP reaction.

**Figure 6 sensors-23-09112-f006:**
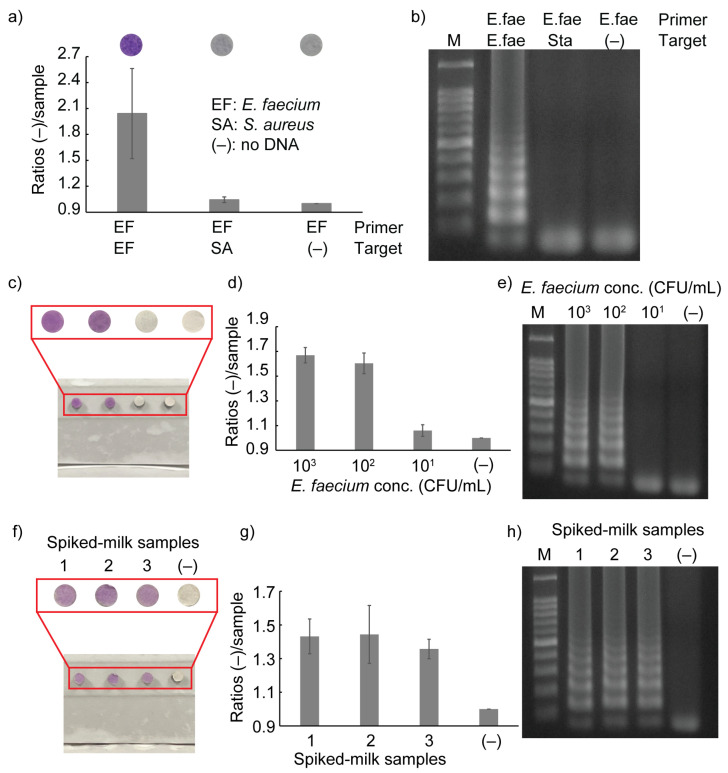
Results showing (**a**) selectivity test and (**b**) electrophoresis. (**c**) The results of the sensitivity test performed using a paper device. (**d**) A graph showing the statistical analyses of the sample containing 10^1^–10^3^ CFU/mL *E. faecium*. (**e**) Electrophoresis results displaying the sensitivity performance. (**f**) Detection of *E. faecium* in milk using the paper device. (**g**) A graph showing the statistical analyses of *E. faecium* detected in the milk sample using the paper device. (**h**) Electrophoresis results obtained when a spiked milk sample was used.

**Table 1 sensors-23-09112-t001:** Primer sequences used for *E. faecium* detection.

Target Gene	Primer Name	Primer Sequence (5′–3′)
*esp* gene (*Enterococcus faecium*)	LB	TGATGTTGACACAACAGTTAAGGG
	F3	CCAGAACACTTATGGAACAG
	B3	GTTGGGCTTTGTGACCTG
	FIP	CGTGTCTCCGCTCTCTTCTTTTTATTTGCAAGATATTGATGGTG
	BIP	ATCGGGAAACCTGAATTAGAAGAAGAACTCGTGGATGAATACTTTC

## Data Availability

Data are contained within the article.

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
