# Peer review of "Pipette-Free and Fully Integrated Paper Device Employing DNA Extraction, Isothermal Amplification, and Carmoisine-Based Colorimetric Detection for Determining Infectious Pathogens"

_sensors, 2023, doi:10.3390/s23229112_

Round 1
Reviewer 1 Report
Comments and Suggestions for Authors
In this submitted manuscript (sensors-2691120), Dr. Lee and Hanh reported their findings in developing a paper-based device for the detection of infectious pathogens. Compared with the other detecting strategy (immunological assays method), this molecular diagnostics way used specific genetic information of pathogens for testing to avoid the issues of false negative results, which has been reported for the immunological assays. Also, in this developed paper-based device, the loop-mediated isothermal amplification (LAMP) was adopted because it features fewer enzymes required, compared to the recombinase polymerase amplification (RPA) approach. What’s more, the colorimetric detection was applied for the LAMP result readout, as it is easy to operate, and has high sensitivity and selectivity.
The authors then carried out experiments and measurements to evaluate the application of this paper-based device, from the aspects of DNA extraction, amplification, and detection. The parameters of detection set-ups were also optimized to realize the best results. It proved that the developed new detecting method has the advantages of easy operation, good compatibility, and high selectivity.
It is important to develop an efficient and accurate method to detect infectious pathogens, as it would have a great influence on human healthcare. The experiments were well-designed and the results were fully discussed. The figures and graphs have clearly shown the setup of this device and experimental results.
The colorimetric detection methods are advantageous compared with conventional strategies. Some good references reporting colorimetric sensing can be considered to cite along with ref 25-27 in the Introduction section: https://doi.org/10.1039/C8QO00963E and doi.org/10.1021/acssensors.3c00287. Also, the language needs to be improved further to correct some grammatical errors found in the manuscript.
Minor modifications are needed to improve the language.
Reviewer 2 Report
Comments and Suggestions for Authors
Accept after minor revision.
Comments:
A pipette-free paper device was designed and prepared to identify E. faecium pathogen based on carmoisine. The threshold value of infectious pathogens can be visual detected and the process is relatively simply, so it has the potential for point-of-care testing. Several suggestions are listed below.
1. In section 3.1 DNA extraction performance, authors investigated the impact of NaOH and SDS concentrations on DNA extraction, the corresponding gel electrophoresis figures should be explained in the text that why the various phenomena was led.
2. The specificity of the prepared device was unclear. Homologous genes are genes that have similar sequences in different species. Due to biological evolution, the function and expression of homologous genes may be different in different species. However, the sequence similarity of homologous genes is very high. Did authors consider the experimental error caused by homologous genes? In other words, is there any other infectious pathogen contained similar targeted gene with E. faecium and does it influence the detection result?

Comments on the Quality of English LanguageMinor editing of English language required.
Reviewer 3 Report
Comments and Suggestions for Authors
1. In the present work, authors have reported an interesting paper based colorimetric sensor for detecting Enterococcus faecium pathogen. The work is interesting and leading to simplify the process of pathogen detection. The only concern is the selectivity / cross reactivity of the developed assay for particular serovars of the tested bacterial species?
2. The author must cite the following study in their literature review part:
Analytical chemistry 84.6 (2012): 2900-2907.
3. Author should discuss the limitations of this study in conclusion part.
